# Nutraceuticals known to promote hair growth do not interfere with the inhibitory action of tamoxifen in MCF7, T47D and BT483 breast cancer cell lines

**Richard Baker** [1], **Giorgio Dell'Acqua**[2], **Aleksander Richards**[2]*, **M. Julie Thornton**[1]*

1 Centre for Skin Sciences, Faculty of Life Sciences, University of Bradford, Bradford, United Kingdom,
2 Nutraceutical Wellness, Inc. New York, NY, United States of America

* aleksander@nutrafol.com (AR); m.j.thornton@bradford.ac.uk (MJT)

## Abstract

**Data Availability Statement:** Data relevant to this study are available from OSF at doi:10.6084/m9.figshare.24926304.

### Background

Hair loss/thinning is a common side effect of tamoxifen in estrogen receptor (ER) positive breast cancer therapy. Some nutraceuticals known to promote hair growth are avoided during breast cancer therapy for fear of phytoestrogenic activity. However, not all botanical ingredients have similarities to estrogens, and in fact, no information exists as to the true interaction of these ingredients with tamoxifen. Therefore, this study sought to ascertain the effect of nutraceuticals (+/- estrogen/tamoxifen), on proliferation of breast cancer cells and the relative expression of ERα/β.

### Methods

Kelp, Astaxanthin, Saw Palmetto, Tocotrienols, Maca, Horsetail, Resveratrol, Curcumin and Ashwagandha were assessed on proliferation of MCF7, T47D and BT483 breast cancer cell lines +/- 17β-estradiol and tamoxifen. Each extract was analysed by high performance liquid chromatography (HPLC) prior to use. Cellular ERα and ERβ expression was assessed by qRT-PCR and western blot. Changes in the cellular localisation of ERα:ERβ and their ratio following incubation with the nutraceuticals was confirmed by immunocytochemistry.

### Results

Estradiol stimulated DNA synthesis in three different breast cancer cell lines: MCF7, T47D and BT483, which was inhibited by tamoxifen; this was mirrored by a specific ERa agonist in T47D and BT483 cells. Overall, nutraceuticals did not interfere with tamoxifen inhibition of estrogen; some even induced further inhibition when combined with tamoxifen. The ERα:ERβ ratio was higher at mRNA and protein level in all cell lines. However, incubation with nutraceuticals induced a shift to higher ERβ expression and a localization of ERs around the nuclear periphery.

**Funding:** MJT received funding from Nutraceutical Wellness, Inc. New York, USA for this study. YES - GD'A contributed to the design of the study and manuscript draft. AR provided feedback on the manuscript.

**Competing interests:** This study was supported by Nutraceutical Wellness, Inc. New York, USA This does not alter our adherence to PLOS ONE policies on sharing data and materials.

**Abbreviations:** ER, estrogen receptor; ERM, selective estrogen receptor modulator; TAM, tamoxifen; E2, 17b-estradiol, K = Kelp; Ast, Astaxanthin; SP, Saw Palmetto; T, Tocotrienols; M, = Maca; H, Horsetail; R, Resveratrol; C, Curcumin; Ash, Ashwagandha; AC, Alternative Curcumin; MRR, mitochondrial retrograde response; NAM, nucleus associated mitochondria; ERB-041, ERb agonist; PPT, ERa agonist.

## Conclusions

As ERα is the key driver of estrogen-dependent breast cancer, if nutraceuticals have a higher affinity for ERβ they may offer a protective effect, particularly if they synergize and augment the actions of tamoxifen. Since ERβ is the predominant ER in the hair follicle, further studies confirming whether nutraceuticals can shift the ratio towards ERβ in hair follicle cells would support a role for them in hair growth. Although more research is needed to assess safety and efficacy, this promising data suggests the potential of nutraceuticals as adjuvant therapy for hair loss in breast cancer patients receiving endocrine therapy.

## Introduction

For women the lifetime risk of developing breast cancer is 1 in 8. Breast cancer is a heterogeneous disease, with several subtypes classified by specific molecular characteristics. Approximately 80% of all breast cancers express the estrogen receptor (ER) [1]. While there are two different nuclear ERs, namely ERα and ERβ, the principal driver of ER-positive breast cancers is ERα. Tumours positive for ERa comprise two main molecular classifications; luminal A (HER2 negative) with low Ki67 expression, or luminal B, (HER2 positive or negative), with high Ki67 expression. For these subtypes, therapies to block ERα signaling are key to endocrine treatment, hence tamoxifen, an agonist of ERa remains the gold standard treatment [1]. Currently, ERβ is not a diagnostic marker, or targeted in breast cancer management. However *in vitro* studies point to an inhibitory role for ERβ in terms of cell proliferation [2], migration, and invasiveness [3, 4], and induction of autophagy [5, 6].

Cell lines established from breast tumours, continue to provide the principal experimental model for cancer research, although significant differences exist between *in vitro* cell lines and tissue samples [7]. The MCF7 cell line has been propagated for almost 50 years and used extensively in breast cancer research [8]. It is a luminal A breast cancer cell line, isolated from the pleural effusion of a 69-year-old woman with metastatic disease [9] and is ER and progesterone receptor (PR) positive, and HER2 negative. Similarly, BT483 and T47D are also luminal A, ER and PR positive, HER2 negative cell lines [10]. T47D was established from the pleural effusion of a 54-year-old woman with ductal breast carcinoma [11], while in contrast, the BT483 line was isolated from a solid, invasive, ductal breast carcinoma of a 23-year-old woman [12]. All three cell lines have been reported to express high levels of ERα protein, while BT483 and T47D also express high levels of prolactin [13].

Endocrine therapy in the form of tamoxifen, which is a selective estrogen receptor modulator (SERM), is prescribed for premenopausal women with ERα-positive breast cancer. Data from the Adjuvant Tamoxifen: Longer Against Shorter (ATLAS) Collaborative Group suggest that 10 years of tamoxifen treatment, as opposed to 5 years, reduces breast cancer mortality by 50% during the second decade after diagnosis [14]. Therefore, many women are prescribed treatment for up to 10 years. Estrogen modulates many non-reproductive tissues, including the hair follicle [15]. However, in women undergoing anti-estrogen treatment, scalp hair loss/thinning is a reported side effect [16, 17]. Patterned hair loss comparable to androgenetic alopecia, or male pattern hair loss was reported in a clinical study of 112 women undergoing treatment for breast cancer [18]. A larger study of 19,430 patients with endocrine-related cancers in 35 clinical trials found the highest incidence of alopecia (25.4%) was in women undergoing tamoxifen treatment in a phase II clinical trial [19]. For many women, endocrine

therapy induced hair loss is significant, negatively impacting sociocultural status and quality of life. It can lead to non-compliance of therapy and therefore poses a therapeutic challenge in patients with breast cancer.

Nutritional factors play an important role in hair growth and shedding, with current adjuvant therapies for hair thinning encompassing vitamin supplementation e.g., vitamin D, E, C, folic acid [20], or omega-3 fatty acids, which improve hair density, reducing the percentage of resting hair follicles [21]. There is increasing evidence that a plethora of plant-derived nutraceuticals including resveratrol, saw palmetto, maca, curcumin, tocotrienols, ashwagandha, horsetail, astaxanthin, kelp, annurca apple fruits, safflower and ginseng, can have beneficial effects on hair growth [22–28]. Some e.g., saw palmetto are competitive, nonselective inhibitors of 5α-reductase [29], the key enzyme required in the development of androgenetic alopecia [30]. These polyphenols exhibit anti-inflammatory and antioxidant activity, and it may be that their mode of action is a protective one, by reducing oxidative stress in the hair follicle.

While nutraceuticals are available without prescription, physicians are often reluctant to endorse them since potential interactions with endocrine therapy has not been fully elucidated.

One prospective study suggests antioxidants taken during chemotherapy, as well as iron and vitamin B12, may increase the risk of recurrence [31], whilst a recent review suggests Vitamin C and Vitamin E improved survival and progression rates [32], yet there are no randomised studies confirming the outcome of antioxidant supplements in breast cancer patients. However, there are wide-ranging reports that nutraceuticals with beneficial effects on hair growth, including curcumin, tocotrienols, kelp and resveratrol, also exert an anti-proliferative effect on breast cancer cells [24, 33–36]. Whether this is due to tissue selectivity, e.g., antagonistic SERM in breast cancer cells, or due to a higher affinity for ERβ remains to be established. The predominant ER in the human hair follicle is ERβ [37, 38], while in breast cancer cells it is ERα, therefore the relative expression and affinity for ERα and ERb is of significance. There is some evidence that some plant polyphenol nutraceuticals can downregulate ERα while stabilizing the anti-proliferative ERβ, resulting in an altered ERα:ERβ ratio [39].

The most important aspect of supplements that can improve endocrine therapy induced hair loss for women undergoing breast cancer treatment is that they should not mimic the stimulatory effect of estrogen on breast cancer cell proliferation, or that they compete with the antagonistic action of tamoxifen. Therefore, understanding their mechanism of action is key since many women will be prescribed endocrine-directed therapy in the form of tamoxifen. A recent study has correlated breast cancer cell lines with tumours using 4 different molecular datasets, namely, gene expression, copy number variation, DNA exome sequencing mutation, and protein phosphorylation expression, to identify which commonly used cell lines have the highest similarity to breast tumours [10]. By using the total similarity score from these correlations with molecular profiles, this study identified BT483 and T47D as the breast cancer cell lines with the highest similarity to tumours, while MCF7 ranked 17[th].

Therefore, the aim of this study was to compare the effect of nine different nutraceuticals on the MCF7 ER-positive breast cancer cell line, along with two other ER-positive breast cancer cell lines that display the highest similarity to tumours *in vivo*. Breast cancer cell proliferation was assessed in the presence of each nutraceutical, and compared with the effect of 17β-estradiol, a specific ERa agonist, and a specific ERb agonist. Any ability to interfere with the inhibitory effect of tamoxifen on 17β-estradiol-stimulated proliferation in each of the cell lines was also evaluated. In addition, changes in the relative expression and cellular localization of ERα and ERβ in the presence of the combined nutraceuticals was established in each cell line.

## Methods

### Cell culture

The cell lines were obtained from European Collection of Authenticated Cell Cultures (MCF7 and T47D) and American Type Culture Collection (BT483). The cell lines were all routinely cultured in complete RPMI containing 2% FBS (MCF7 and T47D) or 10% FBS (BT483) in a humidified incubator at 37°C, 5% $CO_2$.

### Quantitative RT-PCR

Triplicate confluent T75 flasks of each cell line were cultured in serum free RPMI for 48h before extraction of RNA using the Aurum™ Total RNA mini kit. RNA was quantified, normalised, and assessed for purity using a NanoPhotometer and cDNA was synthesised using the iScript™ Advanced cDNA Synthesis kit. A mastermix was prepared containing SsoAdvanced™ Universal SYBR® Green Supermix, cDNA and primers (PrimePCR SYBR Green Assay, Desalt 200R, human primers, ESR1 & ESR2 Bio-Rad, UK), before running on a StepOnePlus RT-PCR System (Applied BioSystems, US) under the following conditions: activation (95°C, 2 mins, 1 cycle), denaturation (95°C, 5 secs, 40 cycles), annealing /extension (60°C, 30 secs, 40 cycles), melt curve (65–95°C, 0.5°C increments, 5 secs/step, 1 cycle). Results were calculated as gene expression relative to the RPS18 housekeeping gene, calculated via the 2-ΔΔCT (Livak) method in relation to a positive control primer for the gene of interest. A gDNA primer was also amplified to ensure the samples were free from contamination along with a reverse transcription (RT) control and PCR run efficiency control to ensure the RT and PCR reactions ran appropriately.

### Western blot

Triplicate confluent T75 flasks of each cell line were cultured in serum free RPMI culture medium for 48h before protein extraction with RIPA Buffer, containing complete™ Protease Inhibitor Cocktail Tablets (Roche, UK). Cell lysates were briefly sonicated, centrifuged at 14,000 rpm for 1min, the supernatant collected, and protein quantified using the BCA Assay Kit (Pierce, UK). Samples were standardised and diluted in 2x Laemmli Buffer and $dH_2O$, boiled for 5 minutes on a hot block and 30μg loaded onto a precast Any kd gel (Bio-Rad, UK) and SDS-PAGE performed at 120V for 90mins. The protein was transferred to LF-PVDF (Bio-Rad, UK) using the Standard SD program on a Transblot Turbo (Bio-Rad, UK). The LF-PVDF was blocked in 5% BSA/PBS for 90mins before the primary antibodies; mouse anti-estrogen receptor alpha antibody (Bio-Rad, UK) and estrogen receptor beta polyclonal antibody (Thermo Fisher, UK) were added at 1:1000, diluted in 1% BSA/PBS, and incubated overnight at 4°C. Membranes were washed in PBST for 5mins before washing 3X in PBS. Alexa Fluor Donkey Anti Mouse 488/Rabbit 647 secondary antibodies (Invitrogen, UK) were diluted 1:1000 in 1% BSA/PBS. hFAB Rhodamine Anti-GAPDH Primary Antibody (Bio-Rad, UK) was added to the antibody solution at 1:2500 and the membranes incubated in the solution for 90 minutes in the dark, at room temperature on a rocker. Membranes were washed once in PBST then 3X t in PBS and imaged on a Chemidoc MP Imager on 488/555/647 nm channels. Images were quantified using Image Lab.

### Estrogen agonists and antagonists

Tamoxifen, 17b-estradiol, were purchased from Sigma UK, and ERB-041 (ERb agonist) and PPT (ERa agonist) from Tocris UK. They were dissolved in DMSO and diluted in serum free RPMI to their working dilutions tamoxifen (2mM), 17b-estradiol (10nM), ERB-041 (100nM) and PPT (100nM).

## Extract origin and preparation

Kelp powder extract (*Laminaria Digitata*) was sourced from Iceland (Thorvin, USA); Astaxanthin powder (*Haematococcus pluvialis*, 5% Astaxanthin), was sourced from China; Saw Palmetto powder (*Serenoa Repens Fruit*), was sourced from the USA; Tocotrienols liquid extract (*Elaeis Guineensis Palm Fruit*, 20% Tocotrienols), was sourced from Indonesia; Resveratrol powder (*Polygonum Cuspidatum*, Resveratrol 50%), was sourced from China; HorseTail powder (*Equisetum Arvense*), was sourced from China; Ashwaghanda powder (*Withania Somnifera*) was sourced from India; Maca powder (*Lepidium meyenii*) was sourced from Peru; Curcumin powder (Curcuma Longa Rhizome) and Alternative Curcumin powder (Encapsulated Curcumin, Curcuma Longa Rhizome) were sourced from India.

Extracts were dissolved at 100 mg/ml in DMSO by gentle warming and agitation in a ThermoMixer. If the extract did not go into solution fully it was then sonicated, and then passed through a 0.22 µm sterile syringe filter. Extracts were HPLC-MS tested to validate the presence of active ingredients as indicated by manufacturer specification. For combined extract treatments, the top tolerated concentration for each extract, ascertained from the WST-1 cytotoxicity assay (see S1 Table and S2 Fig), was combined, and considered 100% for the purposes of further dilutions.

## WST-1 cytotoxicity assay

Cells were seeded into 96 well microplates at a density of $5 \times 10^3$ cells/well in and incubated for 24h before incubating with each extract in serum free medium at 100, 50, 10, 1 and 0.1 µg/ml, or 0.1% DMSO (vehicle control) for 48 and 72 hours. Then the medium was removed, and cells were incubated in 100 µl WST-1 (Abcam, UK) diluted 1:10 in serum-free medium for 4 hours before measuring absorbance on a Tecan Infinite plate reader at 450 nm.

## BrdU incorporation proliferation assay

Cells were seeded into 96 well microplates at a density of $7 \times 10^3$ cells/well and incubated for 24h before incubating with extracts diluted in serum free medium, or 0.1% DMSO (vehicle control) for 24h, before 20µl of BrdU reagent (Sigma, UK) was added and they were incubated for a further 24h. The cells were fixed for 30mins using the FixDenat solution provided, before incubating for 90mins with the anti BrdU-POD solution at room temperature. The cells were washed 3X with PBS and once with $dH_2O$ and incubated with the substrate solution for 30mins before the absorbance was read on a plate reader (Infinite M200, Tecan, Switzerland) at 370 nm.

## Alamar blue proliferation assay

Cells were seeded into 96 well microplates at a density of $5 \times 10^3$ cells/well and incubated for 24h, before incubating with extracts diluted in serum free medium or 0.1% DMSO (vehicle control) for 48h. The medium was removed, and the cells were incubated in 100µl Alamar Blue (Invitrogen, UK) diluted 1:10 in serum-free medium for 4h in the dark before the fluorescence was read on a Tecan Infinite plate reader at excitation 560 nm, emission 590nm.

## Immunocytochemistry

Cells were seeded into 8 well chamber slides at a density of $1 \times 10^4$ cells/well and cultured for 24h and then transferred to serum free medium for 48h before fixing in ice-cold methanol for 15 minutes. To assess changes in estrogen receptor expression cells were incubated with the combined extracts diluted in serum free medium for 4h and 24h before fixing. Cells were air

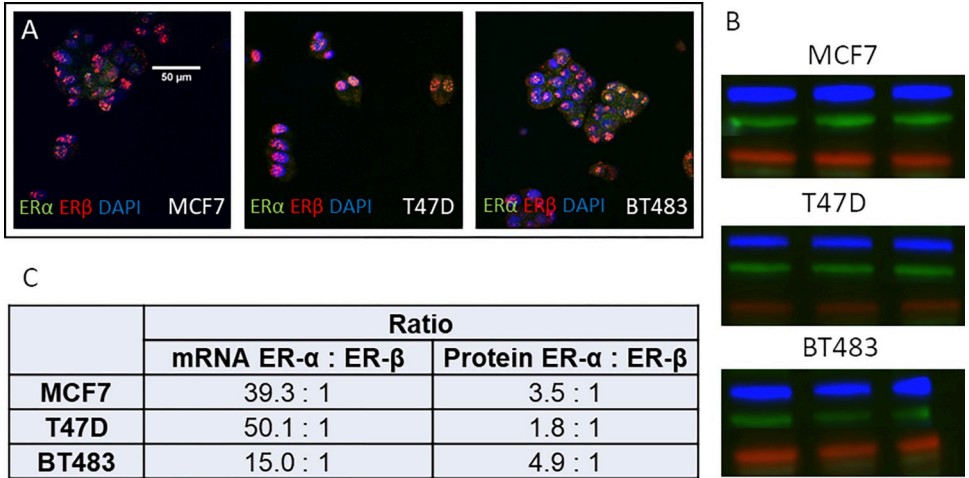

**Fig 1. Cellular localization of ERα:ERβ and quantification of ERα:ERβ mRNA and protein expression in MCF7, T47D & BT483 cells.** (A) Immunocytochemistry: ERα = green, ERβ = red, DAPI = blue, co-localisation = yellow (B) Western blots: ERα = blue, ERβ = green, GAPDH = red. Full western blot image available in S1 Fig. (C) Quantification of mRNA transcripts by qRT-PCR (n = 3 per cell line) and protein expression by quantification of Western blot band density (n = 3 per cell line).

dried, blocked with 10% donkey serum in phosphate buffered saline (PBS) for 1h before incubating with the primary antibody recombinant anti-estrogen receptor alpha antibody and anti- estrogen receptor beta antibody (Abcam, UK), both diluted 1:200 in 1% donkey serum, overnight at 4°C. Cells were washed 3X in PBS before incubating with donkey anti-rabbit alexa fluor 488 and donkey anti-mouse alexa fluor 594 (Invitrogen, UK) secondary antibodies, diluted 1:200 in 1% donkey serum at room temperature for 90mins. Cells were washed 3X in PBS before mounting with Vectashield Mounting Medium (Vector Labs, UK). The slides were imaged on a Zeiss LSM 510 Confocal Microscope. 10 images were then quantified using Image J. Nuclei were outlined using thresholding and region of interest (ROI) and the staining intensity quantified in total and within nuclei, on a pixels per cell basis.

## Results

### All cell lines expressed higher levels of ERa compared to ERb

All cell lines expressed ERa and ERb at the transcript and protein level (Fig 1). However, both mRNA and protein expression of ERa was considerably higher in all three cell lines compared to ERb, although the ratio differed between the cell lines. The ERα:ERβ mRNA ratio was highest in the T47D cells (50:1) and lowest in BT483 (15:1). The difference in protein levels was not as striking, although ERa was still the predominant receptor. In this instance ERa was more highly expressed in the BT483 cell line (4.9:1) and the lowest in the T47D cell line (1.8:1).

### Estrogen stimulation of breast cancer cell proliferation is inhibited by tamoxifen

Incubation with 10nM 17b-estradiol significantly stimulated cell proliferation in all three cell lines (Fig 2A–2C). The most responsive was the BT483 cell line (101%, p<0.0001), followed by T47D (34%, p<0.0001) and then MCF7 (17%, p<0.0001). Stimulation was inhibited by tamoxifen (2-5mM), and although the degree of inhibition was variable between the three cell lines, proliferation was significantly (p<0.0001) reduced when compared to the effect of 17b-

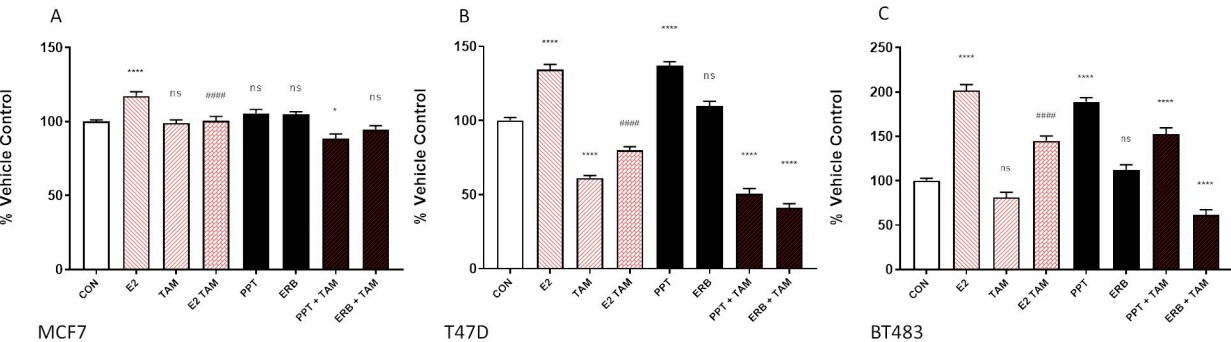

**Fig 2. Cell proliferation is stimulated by 17b-estradiol and an ERa agonist which is negated by tamoxifen: Incorporation of BrdU by breast cancer cells MCF7, T47D and BT483 as a percentage of the vehicle control (+/- SEM of 3 separate assays with 8 experimental replicates).** CON = vehicle control (0.1% DMSO); E2 = 17b-estradiol (10nM); TAM = tamoxifen (2μM); PPT (100nM), ERB = ERB-041 (100nM) Cells were incubated for 48 hours. ANOVA (Dunnett's multiple comparison) statistical significance is displayed on the graph above the bars (* denotes significance vs CON. # denotes significance vs E2.). *$p \leq 0.05$, **$p \leq 0.01$, ***$p \leq 0.001$, ****$p \leq 0.0001$, ns non-significant.

estradiol (Fig 2A–2C). In MCF7 cells, tamoxifen reduced 17b-estradiol stimulation back down to control levels (Fig 2A), while only a partial, yet significant, inhibition, was seen in the BT483 cells (Fig 2C). Tamoxifen alone appeared to have an inhibitory effect on the T47D cells (Fig 2B). The ER-α agonist stimulated a similar response to 17b-estradiol in T47D and BT483 cells, which was similarly negated by tamoxifen. The ER-β agonist had no effect on MCF7 or BT483 cells, although induced a slight stimulation in T47D cells. This was negated in the presence of tamoxifen. Interestingly, tamoxifen in the presence of the ERb agonist significantly reduced proliferation to below basal levels in the BT483 cell line (Fig 2C).

## Confirmation of nutraceutical purity and non-toxic concentrations in vitro

HPLC-MS analysis of the extracts diluted in DMSO confirmed the presence of the active ingredients as indicated by the manufacturer in their specifications.

A cytotoxicity dose response assay on each of the three different cell lines confirmed the maximum tolerated dose for each individual nutraceutical (S1 Table and S2 Fig). All were generally well tolerated by all the cell lines with cytotoxicity only observed at the highest concentrations. The bioavailability and pharmacokinetic data for these extracts is limited. However, all appear to have low bioavailability e.g., a 500mg daily supplement of resveratrol results in plasma levels of 70ng/ml [40] while circulating levels of tocotrienols are approximately 1mg/l following oral supplementation with 400mg [41]. Therefore, since the highest tolerated dose is likely to be supraphysiological we tested three concentrations: the maximum tolerated dose, 10-fold lower and 100-fold lower for all extracts. The concentrations for each extract are indicated in S1 Table and used for all subsequent assays.

## Nutraceuticals did not stimulate breast cancer cell proliferation at lower concentrations

The ability of the following nutraceuticals: kelp, astaxanthin, saw palmetto, tocotrienol, maca, horsetail, resveratrol, curcumin, ashwagandha, alternative curcumin, to stimulate proliferation of all three breast cancer cell lines was assessed either on an individual basis (at 3 different concentrations–see S1 Table), or in combination; extract 1 (all nutraceuticals except alternative curcumin), extract 2 (all nutraceuticals except curcumin). None of the nutraceuticals stimulated cell proliferation, when assessed by the incorporation of BrdU (Fig 3A and 3B), or with the Alamar Blue assay (Fig 3D and 3E) in MCF7 or T47D cells. The BT483 cell line appeared

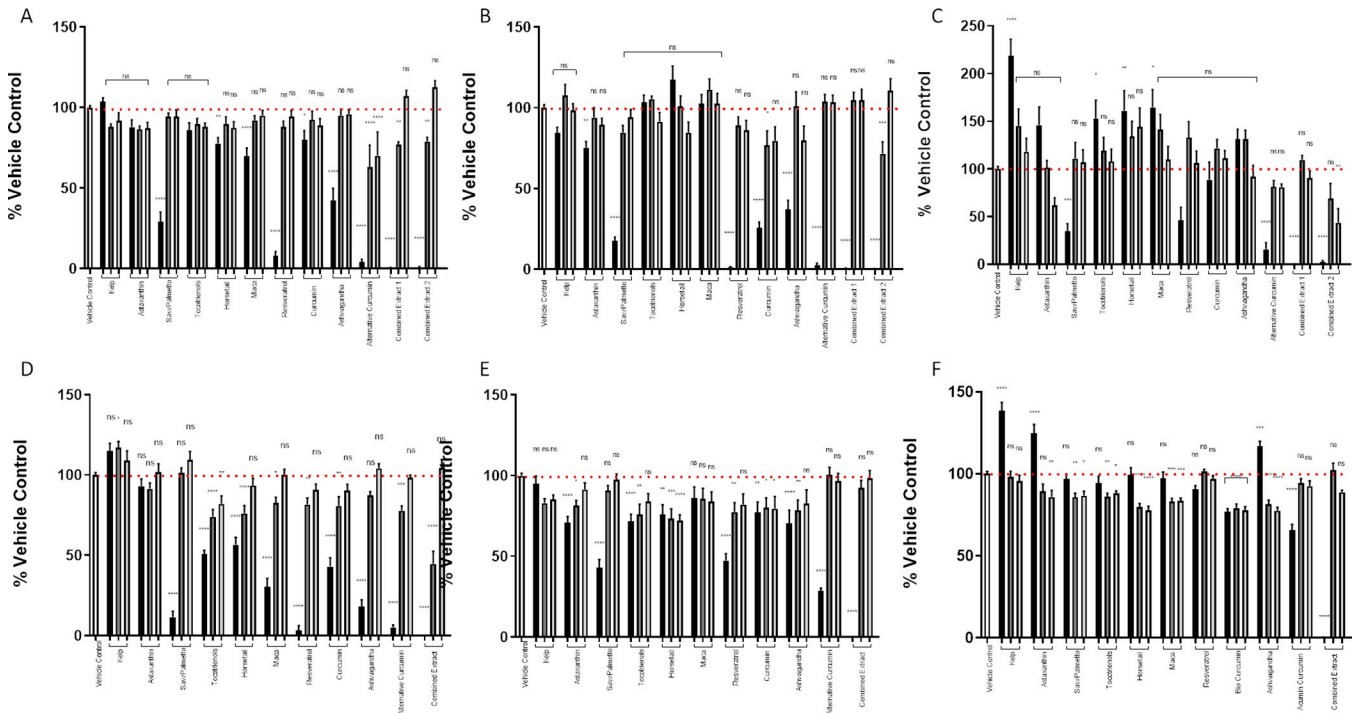

**Fig 3.** Nutraceuticals do not stimulate breast cancer cell proliferation at lower concentrations: Cellular proliferation assessed by incorporation of BrdU by breast cancer cells MCF7 (A), T47D (B), BT483 (C) as a percentage of the vehicle control. Cellular proliferation assessed by Alamar Blue of MCF7 (D), T47D €, BT483 (F) as a percentage of the vehicle control (+/- SEM of 3 separate assays). Vehicle control (0.1% DMSO), Kelp (black bar = 50 µg/ml, dark grey = 5 µg/ml, light grey = 0.5µg/ml), Astaxanthin (black bar = 10 µg/ml, dark grey = 1 µg/ml, light grey = 0.1 µg/ml), Saw Palmetto (black bar = 50 µg/ml, dark grey = 5 µg/ml, light grey = 0.5 µg/ml), Tocotrienols (black bar = 10 µg/ml, dark grey = 1 µg/ml, light grey = 0.1 µg/ml), Maca (black bar = 10 µg/ml, dark grey = 1 µg/ml, light grey = 0.1 µg/ml), Horsetail (black bar = 10 µg/ml, dark grey = 1 µg/ml, light grey = 0.1 µg/ml), Resveratrol (black bar = 10 µg/ml, dark grey = 1 µg/ml, light grey = 0.1 µg/ml), Curcumin (black bar = 1 µg/ml, dark grey = 0.1 µg/ml, light grey = 0.01 µg/ml), Ashwagandha (black bar = 100 µg/ml, dark grey = 10 µg/ml, light grey = 1 µg/ml), Alternative Curcumin (black bar = 10 µg/ml, dark grey = 1 µg/ml, light grey = 0.1 µg/ml) Combined Extract 1 = All extracts combined except Alternative Curcumin (black bar = 25%, dark grey = 10%, light grey = 1%), Combined Extract 2 = All extracts combined except Curcumin (black bar = 25%, dark grey = 10%, light grey = 1%). Cells were incubated with the nutraceuticals for 48 hours. ANOVA (Dunnett's multiple comparison) statistical significance is displayed on the graph above the bars (* denotes significance vs vehicle control.) *$p \leq 0.05$, **$p \leq 0.01$, ***$p \leq 0.001$, ns non-significant.

to be more sensitive and at the very highest concentration kelp (50mg/ml) stimulated proliferation in both assays (Fig 3C and 3F). At their highest concentrations, horsetail (10mg/ml), maca (10mg/ml) and tocotrienol (10mg/ml), stimulated BrdU incorporation, although this was not replicated in the Alamar blue assay. Conversely, the highest tolerated doses of astaxanthin (10mg/ml) and ashwagandha (100mg/ml) did not stimulate BrdU incorporation, but proliferation was increased in the Alamar blue assay. However, when combined with the other compounds at their highest doses, this effect was ablated and became inhibitory. In contrast, none stimulated proliferation of MCF7 or T47D cells and in contrast some had an inhibitory effect (Fig 3, 3A, 3B, 3D and 3E).

## Nutraceuticals did not interfere with the inhibitory effect of tamoxifen on 17b-estradiol stimulated breast cancer cell proliferation at lower concentrations

Nine different nutraceuticals were assessed individually or in combination on their ability to interfere with the tamoxifen inhibition of stimulation by 17b-estradiol, using two different

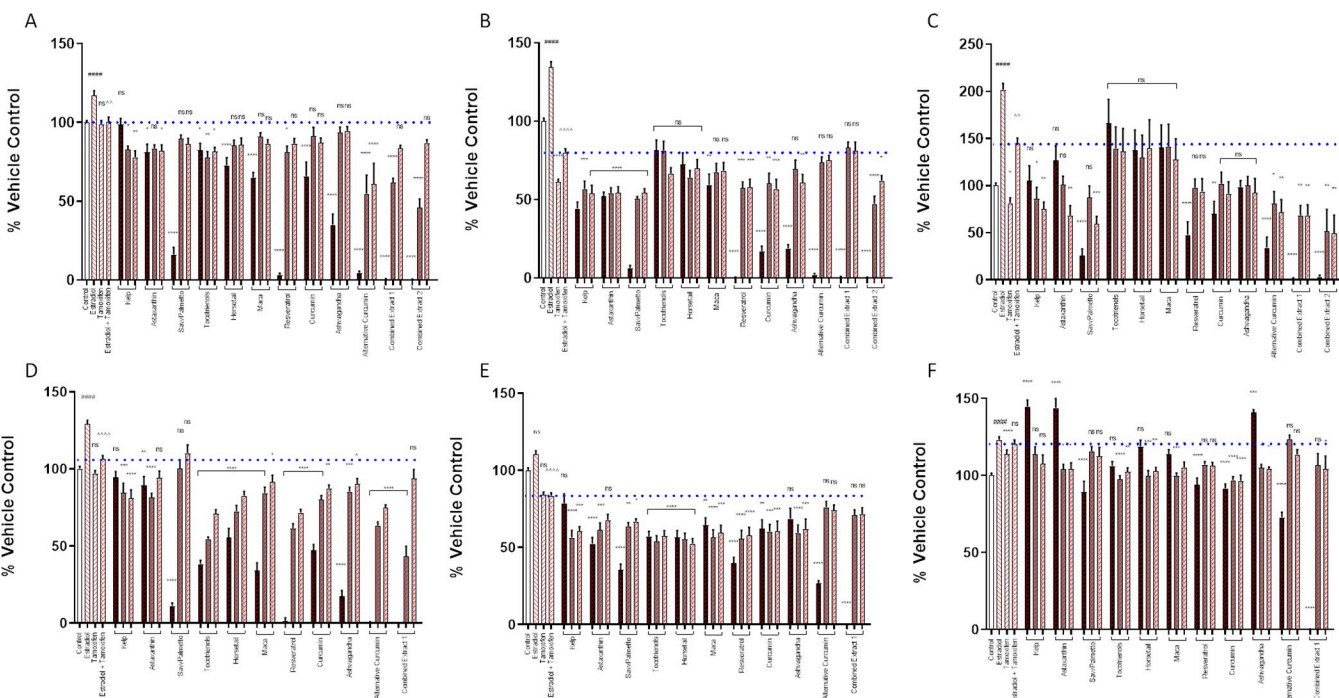

**Fig 4. Nutraceuticals do not interfere with the inhibitory effect of tamoxifen on cell proliferation stimulated by 17b-estradiol at lower concentrations.** Cellular proliferation assessed by incorporation of BrdU by breast cancer cells MCF7 (A), T47D (B), BT483 (C) as a percentage of the vehicle control. Cellular proliferation assessed by Alamar Blue of MCF7 (D), T47D (E), BT483 (F) as a percentage of the vehicle control (+/- SEM of 3 separate assays): Vehicle control (0.1% DMSO); E2 = 17b-estradiol (10nM); Tam = tamoxifen (2µM); Kelp (black bar = 50 µg/ml, dark grey = 5 µg/ml, light grey = 0.5µg/ml), Astaxanthin (black bar = 10 µg/ml, dark grey = 1 µg/ml, light grey = 0.1 µg/ml), Saw Palmetto (black bar = 50 µg/ml, dark grey = 5 µg/ml, light grey = 0.5 µg/ml), Tocotrienols (black bar = 10 µg/ml, dark grey = 1 µg/ml, light grey = 0.1 µg/ml), Maca (black bar = 10 µg/ml, dark grey = 1 µg/ml, light grey = 0.1 µg/ml), Horsetail (black bar = 10 µg/ml, dark grey = 1 µg/ml, light grey = 0.1 µg/ml), Resveratrol (black bar = 10 µg/ml, dark grey = 1 µg/ml, light grey = 0.1 µg/ml), Curcumin (black bar = 1 µg/ml, dark grey = 0.1 µg/ml, light grey = 0.01 µg/ml), Ashwagandha (black bar = 100 µg/ml, dark grey = 10 µg/ml, light grey = 1 µg/ml), Alternative Curcumin (black bar = 10 µg/ml, dark grey = 1 µg/ml, light grey = 0.1 µg/ml) Combined Extract 1 = All extracts combined except Alternative Curcumin (black bar = 25%, dark grey = 10%, light grey = 1%), Combined Extract 2 = All extracts combined except Curcumin (black bar = 25%, dark grey = 10%, light grey = 1%). Cells were incubated with the nutraceuticals for 48 hours in the presence of E2 and tamoxifen. ANOVA (Dunnett's multiple comparison) statistical significance is displayed on the graph above the bars (* denotes significance vs E2 + TAM. # Denotes significance vs vehicle control. ^ Denotes significance vs E2 treated control). *p ≤ 0.05, **p ≤ 0.01, ***p ≤ 0.001.

methods for quantifying cell proliferation. The nutraceutical extracts were incubated in combination with 17b-estradiol and tamoxifen for 48h. None interfered with the inhibitory effect of tamoxifen on BrdU incorporation (Fig 4A–4C) at any of the concentrations used. Even at the lowest concentration, several further augmented the tamoxifen induced inhibition, although this varied between the different cell lines. In MCF7 cells, kelp, astaxanthin and alternative curcumin acted synergistically with tamoxifen to significantly reduce proliferation compared to tamoxifen alone (Fig 4A). Kelp and astaxanthin also had a synergistic effect in T47D cells, along with saw palmetto, resveratrol, ashwagandha, and curcumin, although alternative-curcumin had no effect (Fig 4B. Similar synergy was apparent in the BT483 cell line, in the presence of kelp, astaxanthin, saw palmetto and alternative curcumin inducing further inhibition in the presence of tamoxifen (Fig 4C). Similar results were seen with the Alamar blue assay in MCF7 and T47D cells (Fig 4D and 4E). However, at the very highest concentrations of kelp (50mg/ml), astaxanthin (10mg/ml) and ashwagandha (100mg/ml) there was a stimulation of proliferation in the BT483 cell line as measured by Alamar blue (Fig 4F), which was not seen in the BrdU assay. However, when the nutraceuticals were combined this effect was ablated.

**Table 1. Shift in ratio of ERα:ERβ in the presence of combined nutraceuticals.** Percentage change in ERα:ERβ compared to the vehicle control when cells were treated for 48 hours with the combined extracts. The total amount of ERα & ERβ staining was quantified from randomly selected microscopic fields (n = 5) and calculated as total staining present per nuclei. EXT1 = All extracts combined except AC, EXT2 = All extracts combined except C.

| Percentage Reduction in ERα: ERβ | MCF7 | T47D | BT483 |
|---|---|---|---|
| EXT 1 10% | 75.9% | 129.1% | 49.2% |
| EXT 1 1% | 66.7% | 80.2% | 159.6% |
| EXT 2 10% | 82.4% | -29.9% | 54.0% |
| EXT 2 1% | 72.8% | -49.9% | 51.2% |

## Nutraceuticals induced a shift in the ratio of ERα:ERb in breast cancer cells reducing the relative expression of ERα and increasing the relative expression of ERb

Incubation of breast cancer cells with the combined nutraceuticals, except alternative-curcumin (extract 1), induced a shift in the ratio of ERα:ERb protein expression in favour of ERb (Table 1). The biggest change was in BT483 cells where a concentration of 1% stimulated an increase in the ratio of ERb to ERa by 159%. In contrast, incubation with the combined nutraceuticals without curcumin (extract 2) only increased the ratio of ERb in MCF7 and BT483 cells. In the T47D cells the percentage change in the ratio of ERα:ERb favoured an increased expression of ERa (Table 1).

Following incubation for 48h with combined extracts of the nutraceuticals (extract 1 and extract 2), a shift in the cellular localisation of both ERs from a general nuclear/cytoplasmic localisation to a specific area around the periphery of the nucleus was induced (Fig 5). This was observed in all three cell lines but was most prominent in MCF7 cells.

## Discussion

Cell lines established from breast tumours, continue to provide the principal experimental model for cancer research, although significant differences exist between *in vitro* cell lines and

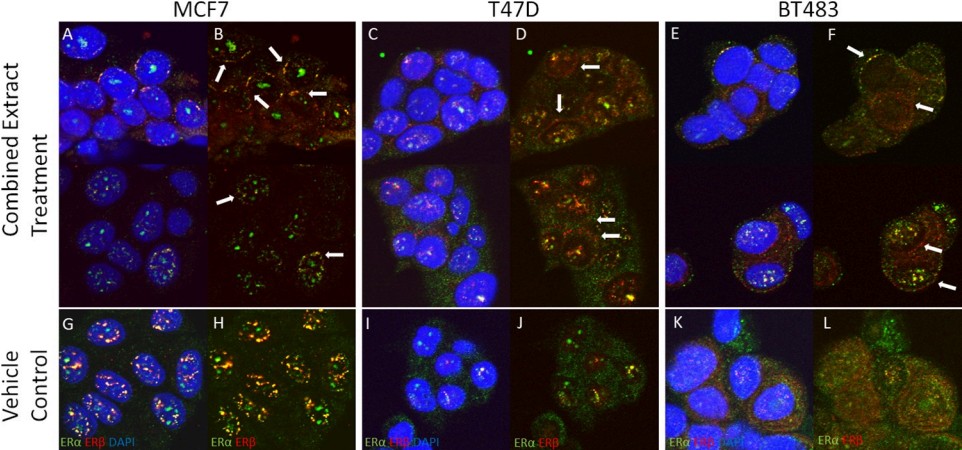

**Fig 5. Cellular localization of ERα:ERβ in presence of combined nutraceuticals.** Cellular localization of ERα:ERβ in presence of combined nutraceuticals. Note the distribution of ERs around the nuclear periphery. MCF7 cells treated with 10% Combined Extract (without Curcumin) with DAPI overlay (A) and without DAPI (B) and vehicle control (G, H). T47D cells treated with 10% Combined Extract (without Alternative Curcumin) with DAPI overlay (C) and without DAPI (D) and vehicle control (I, J). BT483 cells treated with Combined Extract (without Alternative Curcumin) with DAPI overlay (E) and without DAPI (F) and vehicle control (K, L) ERα = green, ERβ = red, Magnification = x630.

tissue samples [7]. The ER positive MCF7 cell line has been propagated for almost 50 years and used extensively in breast cancer research [8]. A recent study correlating breast cancer cell lines with tumours using molecular datasets, have identified BT483 and T47D as the ER positive cell lines which have the highest similarity to breast tumours [10]. Our study has confirmed that all three cell lines have high ERα and low ERβ expression, although the ratio varies at both the transcriptome and protein level (Fig 1). A previous study also reported much higher expression of ERa compared to ERb in MCF7 (93:1) and T47D (23:1) cells by quantifying the proteins via mass spectrometry analysis of 2D-gel spots [42]. This study also compared the proteomic profile of T47D and MCF7 cells, highlighting over 164 proteins were differentially expressed. Proteins with functions in cell proliferation and anti-apoptosis were more highly expressed in T47D cells, while proteins involved in the repression of transcription and regulation of apoptosis were more dominant in MCF7 cells. Such differences further substantiate the heterogeneity of breast cancer and underpin the need to confirm responses in more than one cell line.

In terms of stimulation by 17b-estradiol, the BT483 cell line was the most responsive in our study. Following a 48h incubation with 10nM 17b-estradiol DNA synthesis had increased by 101%, (Fig 2). In comparison, the stimulation of T47D and MCF7 cells by 17b-estradiol was more modest at 34% and 17% respectively, which is in line with previous studies [43]. Under basal conditions BT483 cells grow more slowly [13], so the greater response to 17b-estradiol observed in our study may be a result of their higher relative expression of ERa to ERb protein. It may also be due having a molecular profile with a much higher similarity to *in vivo* tumours [10]. Many of the commonly used breast cancer cell lines are derived from pleural effusions, including MCF7 and T47D, whereas BT483 is derived from a solid breast carcinoma [12].

Incubation with PPT, which is a selective ERa modulator with no affinity for ERb [44] significantly stimulated proliferation of both T47D and BT483 cells (Fig 2), although it did not significantly increase MCF7 proliferation. Again, this may reflect the higher response of T47D and BT483 cells to 17b-estradiol and their greater similarity to breast tumours compared to MCF7 cells [10]. Incubation with ERB-104, a preferential modulator of ERb with 220-fold selectivity for ERb over ERa [45] did not stimulate proliferation of MCF7 or BT483 cells (Fig 2), although there was a small stimulation in T47D cells. This may be attributed to ERa:ERb expression, particularly since the T47D cells had the highest ratio at the transcriptional level and the lowest ratio at the protein level (Fig 1). The concentration of ERB-104 used in the current study was 10-fold higher than that of 17b-estradiol. A study of botanical estrogens with a higher affinity and selectivity for ERβ reported that at high concentrations they were able to stimulate cell proliferation via ERα, highlighting that their concentration and the cellular ERα:ERβ ratio significantly modulated their subsequent biological effects [46].

Phytoestrogens share some structural similarities to 17b-estradiol, specifically a phenolic hydroxyl A ring. This structure is paramount for interaction with ERα and ERβ ligand binding domains, which permits them to simulate the effects of 17b-estradiol [47]. Since they can modulate agonistic or antagonistic responses, such polyphenolic compounds are deemed to be naturally occurring SERMs [48]. However, several studies have shown that phytoestrogens have a higher affinity for ERb [49, 50]. To explore the response of human breast tumours expressing different ratios of ERa and ERb, Jiang *et al.* [46], compared their effects on MCF7 cells containing only ERα, only ERβ, or both ERα and ERβ. They identified that following preferential binding to ERβ, phytoestrogens induced co-activator recruitment and stimulated chromatin binding to enhance expression of ERβ-regulated genes, thereby demonstrating that ERa:ERb selectivity and binding affinity of phytoestrogens is augmented at the epigenetic level.

In the current study, none of the nutraceuticals tested at any concentration, either alone or in combination, increased cell proliferation in the MCF7 and T47D cell lines in either the

BrdU assay (Fig 3A–3C) or the Alamar blue assay (Fig 3D–3F), while some actually inhibited proliferation. While this was also the case in BT483 cells at the two lower concentrations of nutraceuticals tested, at very high concentrations, namely kelp, horsetail, maca, tocotrienol, astaxanthin and ashwagandha stimulated proliferation as measured by either BrdU or Alamar blue. Only kelp stimulated proliferation in both assays (Fig 3C and 3F). However, when all nutraceuticals were combined this effect was ablated. MCF7 and T47D are the most well-known and characterised ER positive breast cancer cell lines used for *in vitro* studies. We chose to include BT483 cells in this study since they have the highest similarity to tumours *in vivo* [10]. We also demonstrated that BT483 cells had the highest protein expression of ERα in relation to ERb than the other cell lines (Fig 1) and had a greater proliferative response to estradiol than either MCF7 or T47D cells (Fig 2C). The BT483 cells originate from a much younger donor and appear much more sensitive to stimulation than more commonly used breast cancer cell lines and further highlight the heterogeneity of breast cancer cells and emphasise the need to measure effects in more than one cell type.

However, the main aim of this study was to address whether these nutraceuticals are activating estrogen receptors as potential SERMs, and more importantly can they compete with another SERM, in this case tamoxifen? This is of particular significance for women that may be taking even non-phytoestrogenic nutraceuticals to prevent endocrine therapy induced hair loss [24]. Tamoxifen is a non-steroidal triphenylethylene prescribed to pre-menopausal women for the hormonal treatment of ER positive breast cancer. It interacts with both ERs although its affinity for ERa is approximately twice that of ERb [51]. However, any competition with binding of tamoxifen to ERs in breast cancer cells would limit its therapeutic capability. Tamoxifen blocked the stimulation of 17b-estradiol-induced proliferation in all three cells lines demonstrating its effectiveness as an estrogen antagonist (Fig 4A–4F). In MCF7 and T47D cells, tamoxifen reduced 17b-estradiol-induced DNA synthesis back to the basal level, while in BT483 cells, although significantly reduced, it was not fully lowered to basal levels. This is in line with other studies where tamoxifen only partially, but significantly inhibited the stimulatory effect of 17b-estradiol on breast cancer cell lines, including MCF7, T47D, ZR-75-1 and BT474 [52].

None of the nutraceuticals interfered with the inhibitory effect of tamoxifen on 17b-estradiol-induced proliferation in any of the cell lines (Fig 4A–4C), as measured by the BrdU assay. Although variable, several of them further augmented the tamoxifen induced inhibition, reducing proliferation below basal levels indicating a synergistic effect with tamoxifen. Kelp and astaxanthin acted synergistically with tamoxifen to significantly reduce proliferation compared to tamoxifen alone in all three cell lines. Saw palmetto had a synergistic effect in T47D and BT483 cells, while alternative curcumin displayed synergy in MCF7 and BT483 cells, and curcumin in T47D cells. Resveratrol and ashwagandha were also effective in T47D cells (Fig 4A–4C). Although in MCF7 and T47D cells this was further corroborated by the Alamar blue assay (Fig 4D and 4E), in the BT483 cells there was a stimulation by kelp, astaxanthin and ashwagandha, but only at the very highest concentration used. The differences observed between the two assays are likely due to the different techniques of assessing cell proliferation in culture by each assay. The BrdU assay measures the incorporation of BrdU during DNA synthesis, replacing thymidine. Therefore, the measured absorbance correlates directly with the amount of DNA synthesis and the number of proliferating cells present. In contrast, Alamar Blue utilizes the reduction of resazurin by cells, so the amount of absorbance is proportional to the number of living cells and corresponds to cellular metabolic activity. In BT483 cells, the observed stimulation is likely due to an increase in cellular metabolism, that is not necessarily translating into proliferation. Notwithstanding, this study highlights differences in the response of difference breast cancer cells lines, accentuates their heterogeneity and the importance of using a number of different donors.

The synergistic effect of some of the nutraceuticals with tamoxifen suggests they may provide additional beneficial effects. Previously it has been reported that tocotrienols inhibit the proliferation of MCF7 cells, and when combined with tamoxifen reduce proliferation even further [53]. Likewise, while synergy with tamoxifen has not been reported in breast cancer cells, curcumin synergises with tamoxifen in the recovery of $H_2O_2$-induced myocardial apoptosis in ventricular cardiomyocytes of neonatal rats [54]. More recently a study combining novel analogues of resveratrol with tamoxifen demonstrated a synergistic inhibition of the proliferation of breast cancer cell lines, including MCF7 and T47D [55]. Mechanistic studies revealed that in MCF7 and T47D cells, this synergy was due to down regulation of ERα and the oncogene c-Myc.

Higher expression of ERβ is associated with improved 5-year disease free survival and overall survival in ERα breast cancer patients [56]. *In vitro* studies have shown that 17b-estradiol-induced T47D proliferation was no longer observed when ERb expression was increased, demonstrating that the ability of estrogen-like compounds to stimulate cell proliferation are dependent on the cellular ratio of ERa/ERb expression, and whether they have a higher affinity for ERa or ERb [44]. In the current study incubation with the combined nutraceuticals induced a shift in the protein expression of ERa/ERb in favour of ERb (Table 1), suggesting that their effect on reducing cell proliferation may be in part, induced by lower ERa and increased ERb activity. An interesting observation was the change in the cellular localisation of ERs following incubation with combined nutraceuticals, to a specific area around the periphery of the nucleus (Fig 5), which was observed in all three cell lines. Recently it has been demonstrated that cellular redistribution of mitochondria during the mitochondrial retrograde response (MRR) can be facilitated by contact sites with the nucleus; coined nucleus associated mitochondria (NAM) [57]. It has also been reported that mitochondria express ERα and ERβ that regulate oxidative stress originating in the mitochondria [58]. Furthermore, in MCF-7 cells, tamoxifen activates mitochondrial ERb as an antagonist to inactivate manganese superoxide dismutase which upregulates superoxide-induced apoptosis [59]. Further mechanistic studies to determine the impact of nutraceuticals and their combinations on modifying the ERα/ERβ ratio, and their potential role in regulating mitochondrial function are required.

Many breast cancer patients are prescribed tamoxifen, but this may induce endocrine-therapy induced hair loss, which for many women is an unacceptable side-effect resulting in non-compliance of treatment. Several nutraceutical formulations have shown clinical efficacy in addressing hair loss and as such may be sought out by patients experiencing thinning on tamoxifen. The most important feature of supplements used in endocrine-therapy induced hair loss is that ingredients should not mimic 17b-estrogen or compete with the antagonistic action of tamoxifen. The ideal scenario would be an antagonist in breast cancer cells and an agonist in the hair follicle. ERb is the predominant ER in the human hair follicle therefore their potential to shift the ERα/ERβ ratio in favour of ERb in hair follicle cells may be of consequence and warrants further investigation. The hair follicle epidermal matrix cells are the second highest proliferative cells in the human body and susceptible to oxidative stress, therefore the antioxidant mechanism action of these nutraceuticals may also be an important contributing factor to their mechanism of action. While further studies are required to assess safety and efficacy, this encouraging data supports the safety and potential for use of nutraceuticals as adjuvant therapy for hair loss or thinning in breast cancer patients receiving endocrine therapy.

## Supporting information

**S1 Fig. Whole image of western blot from Fig 1.**
(TIF)

**S2 Fig. Cytotoxicity assessment of nutraceuticals on MCF7, T47D and BT483 cells by WST-1.** Breast cancer cells were treated with Kelp (A), Astaxanthin (B), Saw Palmetto (C), Tocotrienols (D), Horsetail (E), Maca (F), Resveratrol (G), Curcumin (H), Ashwagandha (I) and Alternative Curcumin (J) for 48 & 72 hours at concentrations of 0.1–100 μg/ml. Cytotoxicity was then assessed using WST-1 as a percentage of the vehicle control (0.1% DMSO). (TIF)

**S1 Table. Table showing suggested maximum experimental concentrations based upon the cytotoxicity observed in the WST-1 assays.** 2 log-fold dilutions from the maximum tolerated concentration were also tested in subsequent experiments. (DOCX)

# Author Contributions

**Data curation:** Richard Baker.

**Formal analysis:** Richard Baker.

**Funding acquisition:** M. Julie Thornton.

**Methodology:** Richard Baker, Giorgio Dell'Acqua, M. Julie Thornton.

**Supervision:** M. Julie Thornton.

**Writing – original draft:** Giorgio Dell'Acqua, M. Julie Thornton.

**Writing – review & editing:** Aleksander Richards, M. Julie Thornton.

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
