## [Decision Letter · Decision Letter 0]

20 Oct 2023

PONE-D-23-31623Nutraceuticals known to promote hair growth do not interfere with the inhibitory action of tamoxifen in MCF7, T47D and BT483 breast cancer cell linesPLOS ONE

Dear Dr. Baker,

Thank you for submitting your manuscript to PLOS ONE. After careful consideration, we feel that it has merit but does not fully meet PLOS ONE’s publication criteria as it currently stands. Therefore, we invite you to submit a revised version of the manuscript that addresses the points raised during the review process.

We look forward to receiving your revised manuscript.

Kind regards,

Wei Xu

Academic Editor

PLOS ONE

“MJT received funding from Nutraceutical Wellness, Inc. New York, USA for this study.

YES - GD’A contributed to the design of the study and manuscript draft. AR provided feedback on the manuscript.”

“This study was supported by Nutraceutical Wellness, Inc. New York, USA.”

“This study was supported by Nutraceutical Wellness, Inc. New York, USA in collaboration with the Centre for Skin Sciences, University of Bradford, UK.”

“MJT received funding from Nutraceutical Wellness, Inc. New York, USA for this study.

YES - GD’A contributed to the design of the study and manuscript draft. AR provided feedback on the manuscript.”

7. We notice that your supplementary table is included in the manuscript file. Please remove them and upload them with the file type 'Supporting Information'. Please ensure that each Supporting Information file has a legend listed in the manuscript after the references list.

Reviewers' comments:

Reviewer's Responses to Questions

**Comments to the Author**

1. Is the manuscript technically sound, and do the data support the conclusions?

Reviewer #1: No

Reviewer #2: Yes

2. Has the statistical analysis been performed appropriately and rigorously? 

Reviewer #1: No

Reviewer #2: Yes

3. Have the authors made all data underlying the findings in their manuscript fully available?

Reviewer #1: Yes

Reviewer #2: Yes

4. Is the manuscript presented in an intelligible fashion and written in standard English?

Reviewer #1: Yes

Reviewer #2: Yes

5. Review Comments to the Author

Reviewer #1: Considering that alopecia is a common side effect of treatment with tamoxifen, the authors tested the response of main components and extracts of nutraceuticals using three ER-positive breast cancer cells. However, there are still some questions remaining to be addressed.

1. In Figure 3, the authors mentioned that the effects of AST and H on BT483 cells were not statistically significant, but the altered cell viability was nearly 50%. Thus, the BrdU data’s significance is probably to be re-calculated. Moreover, other assays should be applied to make the conclusions more reliable, for example, the EdU assay. The data of the Alamar Blue Proliferation Assay should be included in the manuscript since the authors draw conclusions through this assay.

2. The authors treated breast cancer cell lines with estrogen for 48 hours in this paper, however, short-term treatment, like 15 minutes, is sufficient to activate the proliferative pathway. Besides 48 hours, a short-term treatment should be performed to detect cell proliferation by flow cytometry and BrdU assay.

3. The authors treated cells with combined compounds or extracts. Did they do experiments to figure out whether there are any chemical reactions between these added compounds?

4. Besides location, the expression levels of ERα and ERβ are important to cellular proliferation. Thus, the authors should also test the expression levels of ERα and ERβ in their research.

5. The organization of the figures is not usual, different panels of data in the same figure should be labeled from A to Z.

Reviewer #2: Hair loss/thinning is common side effect of chemotherapy for cancer treatment. To investigate how to reduce the side effect without affect the efficacy is interesting. In the manuscript, R Baker and colleagues measured the effects of several nutraceuticals, known to benefit for hair growth, on ER+ breast cancer cell proliferation with/without tamoxifen treatment. Although there are preliminary data, it is a good trial.

Comments:

1. Figure 4, the authors addressed that nutraceuticals did not interfere with the efficacy of tamoxifen, even in some cell lines synergistic effect was observed. However, they just tested one concentration. To distinguish the effect whether it is additive or synergistic, multiple concentrations of the two components should be tested.

2. “Table 1. Shift in ratio of ERα:ERβ in the presence of combined nutraceuticals” Western blot of ER alpha and beta should be involved in.

3. Figure 5, DAPI staining should be done to show the nucleus.

6. PLOS authors have the option to publish the peer review history of their article (what does this mean?). If published, this will include your full peer review and any attached files.

Reviewer #1: No

Reviewer #2: No

---

## [Author Response · Author response to Decision Letter 0]

19 Dec 2023

Responses to Reviewers (please see attached documents for a copy of this)

We thank the reviewers for their constructive comments and have addressed these point-by-point below. All changes to the manuscript are highlighted in track changes.

Reviewer 1: 

Considering that alopecia is a common side effect of treatment with tamoxifen, the authors tested the response of main components and extracts of nutraceuticals using three ER-positive breast cancer cells. However, there are still some questions remaining to be addressed.

1. In Figure 3, the authors mentioned that the effects of AST and H on BT483 cells were not statistically significant, but the altered cell viability was nearly 50%. Thus, the BrdU data’s significance is probably to be re-calculated. Moreover, other assays should be applied to make the conclusions more reliable, for example, the EdU assay. The data of the Alamar Blue Proliferation Assay should be included in the manuscript since the authors draw conclusions through this assay.

Additional data has been included, a further 2 concentrations (10-fold and 100-fold) higher and the data recalculated to include these additional values with Dunnett’s multiple comparison to capture significant differences normalized against the control values. At the highest concentration, kelp (50�g/ml) does significantly stimulate DNA synthesis, as does horsetail but ONLY in the BT483 cell line (figure 3C). However, when they are used in combination with the other extracts, there is an inhibition of DNA synthesis. The Alamar blue data has now been included (figure 3D to 3F and figure 4D to 4F) – using the same three different concentrations for each nutraceutical as has been tested as in the BrdU assay. Interesting, using this alternative assay, these nutraceuticals did not stimulate proliferation. Conversely astaxanthin and ashwagandha stimulated proliferation in this assay, but again ONLY in the BT483 cell line, and only at the highest concentration (figure 3F). Again, when they are used in combination with the other extracts, this effect is ablated.

2. The authors treated breast cancer cell lines with estrogen for 48 hours in this paper, however, short-term treatment, like 15 minutes, is sufficient to activate the proliferative pathway. Besides 48 hours, a short-term treatment should be performed to detect cell proliferation by flow cytometry and BrdU assay.

ER alpha and ER beta are both nuclear transcription factors and therefore genomic activation of steroid receptors is a slower response than activation of cell surface receptors which utilise rapid second messenger signalling pathways. While it is acknowledged that estrogen can induce rapid signalling cascades through non-genomic pathways, typically the genomic effects of steroid hormones take longer, with a time-lag of at least 2h, with changes in gene expression occurring on the timescale of hours, not minutes. Since we are specifically looking at activation of genomic ER alpha and ER beta in breast cancer proliferation, a short period of 15min would indicate a non-genomic signalling cascade had been activated. Since tamoxifen in breast cancer cells binds to genomic ER-alpha to block the nuclear receptor, a long incubation period is required to establish that these compounds do not interfere with antagonist action of tamoxifen.

3. The authors treated cells with combined compounds or extracts. Did they do experiments to figure out whether there are any chemical reactions between these added compounds?

The aim of this study was to establish that the nutraceuticals did not interfere with the inhibitory effect of tamoxifen on estrogen stimulated DNA synthesis in breast cancer cells (figure 4). This was confirmed by BrdU in all three cell types both individually and when combined. However, ONLY in the BT483 cell line, three compounds, namely kelp, astaxanthin and ashwagandha at very high concentrations stimulated proliferation- but ONLY as measured using the Alamar blue assay. However, when used in combination with the other nutraceuticals this effect was not seen. The differences observed between the two assays are likely due to the different techniques of assessing cell proliferation in culture by each assay. The BrdU assay measures the incorporation of BrdU during DNA synthesis, replacing thymidine. Therefore, the measured absorbance correlates directly with the amount of DNA synthesis and the number of proliferating cells present. In contrast, Alamar Blue utilizes the reduction of resazurin by cells, so the amount of absorbance is proportional to the number of living cells and corresponds to cellular metabolic activity. In BT483 cells, the observed stimulation is likely due to an increase in cellular metabolism, that is not necessarily translating into proliferation. Notwithstanding, this study highlights differences in the response of difference breast cancer cells lines, accentuates their heterogeneity and the importance of using several different donors. This also highlights the importance of using more than one cell line and more than one assay in this type of study.

4. Besides location, the expression levels of ERα and ERβ are important to cellular proliferation. Thus, the authors should also test the expression levels of ERα and ERβ in their research.

The relative expression of mRNA for ERα and ERβ has been measured by qRT-PCR and the protein expression confirmed by Western blot in all three cell lines (figure 1). The full western blot image has been included as supplementary figure 1. This confirms that while ERα is the major estrogen receptor in all three cells lines at both the gens and protein level, the ratio of ERα to ER� varies between the three different cell lines.

5. The organization of the figures is not usual, different panels of data in the same figure should be labeled from A to Z.

Thank you for drawing our attention to this. This has now been rectified and each figure labelled appropriately.

Reviewer 2: 

Hair loss/thinning is common side effect of chemotherapy for cancer treatment. To investigate how to reduce the side effect without affect the efficacy is interesting. In the manuscript, R Baker and colleagues measured the effects of several nutraceuticals, known to benefit for hair growth, on ER+ breast cancer cell proliferation with/without tamoxifen treatment. Although there are preliminary data, it is a good trial.

Comments:

1. Figure 4, the authors addressed that nutraceuticals did not interfere with the efficacy of tamoxifen, even in some cell lines synergistic effect was observed. However, they just tested one concentration. To distinguish the effect whether it is additive or synergistic, multiple concentrations of the two components should be tested.

We have included two further concentrations of nutraceuticals for each cell line -10-fold and 100-fold higher than the original concentration displayed in figures 3 and 4. Each figure now shows 3 different concentrations for each nutraceutical. The effects still appear to be synergistic, although at the highest concentration, some nutraceuticals appear to have a cytotoxic effect on cell metabolism/ DNA synthesis. This was not reflected in the cytotoxicity (WST-1 Cytotoxicity Assay) that was used to establish the maximum dose at which each compound could be used. At the highest tolerated dose as assessed by the WST-1 assay, the BrdU (DNA synthesis) and Alamar blue (metabolic activity) assays may be more sensitive and the concentrations at the limit of what the cells can tolerate.

2. “Table 1. Shift in ratio of ERα:ERβ in the presence of combined nutraceuticals” Western blot of ER alpha and beta should be involved in.

This has been added as a supplementary figure.

3. Figure 5, DAPI staining should be done to show the nucleus.

The corresponding images showing the DAPi staining have now been included in figure 5.

---

## [Editor Report · Decision Letter 1]

28 Dec 2023

Nutraceuticals known to promote hair growth do not interfere with the inhibitory action of tamoxifen in MCF7, T47D and BT483 breast cancer cell lines

PONE-D-23-31623R1

Dear Dr. Baker

We’re pleased to inform you that your manuscript has been judged scientifically suitable for publication and will be formally accepted for publication once it meets all outstanding technical requirements.

Kind regards,

Wei Xu

Academic Editor

PLOS ONE
---

## [Editor Report · Acceptance letter]

14 Feb 2024

PONE-D-23-31623R1 

PLOS ONE

Dear Dr. Baker, 

I'm pleased to inform you that your manuscript has been deemed suitable for publication in PLOS ONE. Congratulations! Your manuscript is now being handed over to our production team.

Kind regards, 

on behalf of

Dr. Wei Xu 

Academic Editor

PLOS ONE